

# Prognostic value of lymphocyte-to-monocyte ratio in patients with endometrial cancer: an updated systematic review and meta-analysis

Zijing Huang[1], Donghua Yang[2] and Congrong Liu[3]

[1] Peking University First Hospital, Beijing, China
[2] College of Urban and Environmental Sciences, Peking University, Beijing, China
[3] Department of Pathology, Third Hospital, School of Basic Medical Sciences, Peking University Health Science Center, Beijing, China

Corresponding author
Congrong Liu,
congrong_liu@hsc.pku.edu.cn

## ABSTRACT

**Background**. Evaluating the risk of metastasis at diagnosis and the likelihood of future recurrence is crucial for the effective management of endometrial cancer (EC). While conventional prognostic indicators hold importance, they often fall short in predicting recurrence, especially in low-risk patients. This study evaluates the prognostic value of the lymphocyte-to-monocyte ratio (LMR) for overall survival (OS), disease-free survival (DFS), and cancer-specific survival (CSS) in EC patients.

**Methods**. Eligible studies that provided pretreatment cutoff values of LMR, hazard ratios (HRs), and 95% confidence intervals (CIs) for OS, DFS, CSS, and progression-free survival (PFS) were included in this meta-analysis. Two independent reviewers collected and evaluated the data, and the quality of the included studies was assessed using the Newcastle Ottawa Quality Assessment Scale (NOS). Statistical analyses were performed using STATA software, and subgroup analyses were conducted by race, sample size, and age to assess the consistency of LMR's prognostic value across different population groups.

**Results**. In this meta-analysis, eight studies were included for OS (1,997 patients) and five studies were included for DFS (1,590 patients). LMR was significantly associated with OS (HR 2.29; 95% CI [1.50–3.51]; $p = 0.0014$), DFS (HR 4.00; 95% CI [1.76–9.07]; $p = 0.0094$), and CSS (HR, 1.58; 95% CI [1.11–2.25]; $p = 0.01$). Subgroup analysis indicated that the prognostic value of LMR for OS was consistent across different races, age groups, and sample sizes. However, the correlation between LMR and DFS was influenced by median age, with younger patients (<60 years) showing a stronger association. Sensitivity analyses confirmed the robustness of these results, and Egger's test showed no significant publication bias.

**Discussion**. LMR serves as a valuable prognostic marker for OS, DFS, and CSS in EC patients. Its predictive power remains significant across diverse population groups, underscoring its potential utility in clinical practice. Biological mechanisms linking inflammation and cancer support the role of LMR in prognosis, given the functions of lymphocytes and monocytes in tumor progression and immune response. These findings suggest that incorporating LMR into current prognostic models could enhance risk stratification for EC patients, particularly for identifying those at higher risk of recurrence despite being classified as low risk by traditional systems. In conclusion,

LMR is a robust, independent prognostic factor for EC, with significant implications for improving patient management and outcomes through better risk stratification.

## INTRODUCTION

Endometrial cancer (EC) originates in the endometrium and is the predominant gynecologic malignancy and a major contributor to cancer-related morbidity and mortality among women in high-income countries worldwide. Although it is more prevalent in developed regions like North America and Europe, the incidence of EC is also on the rise in several developing countries, posing a considerable public health challenge (*Ferlay et al., 2019*; *Siegel et al., 2022*). While the precise etiology of EC is complex and multifactorial, several potential risk factors for which has been established, including elevated estrogen levels, obesity, and certain genetic disorders (*Merritt et al., 2016*; *Raglan et al., 2019*; *Rutanen et al., 1994*). For EC diagnosis, the 2023 FIGO staging system integrates additional prognostic factors, including histological type, molecular classification, lymphovascular space invasion (LVSI), and lymph node metastasis, compared to the 2009 version. This comprehensive approach improves the accuracy of risk assessment and prognosis prediction for EC patients.

New prognostic tools are needed to better identify women at higher risk of mortality. Recently, various potential biomarkers, such as miRNAs and genes that may be associated with the pathogenesis and prognosis of EC have been investigated (*Liu, Lin & He, 2019*; *Wang et al., 2020*). Among these emerging prognostic factors, the lymphocyte-to-monocyte ratio (LMR) can be easily obtained by routine preoperative blood tests and holds promise for predicting the prognosis of cancer patients. This ratio reflects the balance between lymphocytes and monocytes in the blood as a marker for immune response and inflammation in the body, particularly in various tumors, including ovarian, larynx, and EC (*Cichowska-Cwalińska et al., 2023*; *Cong et al., 2020*; *Huszno et al., 2022*; *Jeong et al., 2023*; *Song et al., 2021*). Specifically, studies also indicate an inverse relationship between LMR and cancer outcomes, with lower LMR associated with a higher risk of adverse events, disease severity, and poorer prognosis (*González-Sierra et al., 2023*; *Oksuz et al., 2017*; *Wang et al., 2017*). Recent studies have also highlighted the potential of LMR as a superior diagnostic and prognostic biomarker compared to other indicators in certain cancer types, underscoring its clinical relevance and research value (*Kang et al., 2021*; *Wang et al., 2024*).

Multiple studies have explored the relationship between LMR and clinical outcomes in EC patients, demonstrating its potential prognostic value in predicting survival and recurrence. However, the predictive value of LMR for EC prognosis is still under debate (*Ahn et al., 2022*; *Cummings et al., 2015*; *Eo et al., 2016*; *Holub et al., 2020*; *Song et al., 2023*). As more recent studies are published, the significance of LMR as a predictive factor for EC

has been increasingly realized. Therefore, this study aims to integrate existing research and provide more compelling evidence for the prognostic value of LMR in EC patients.

## MATERIALS & METHODS

### Search strategy

This paper followed the PRISMA guideline (ID number: CRD42024506630). PubMed, EMBASE, Cochrane Library, and WOS were systematically searched for potentially eligible studies until November 2023 without date restrictions. The search strategies in PubMed were: (((((((((((((((((((((Endometrial Neoplasms) OR (Endometrial Neoplasm)) OR (Neoplasm, Endometrial)) OR (Neoplasms, Endometrial)) OR (Endometrial Carcinoma)) OR (Carcinoma, Endometrial)) OR (Carcinomas, Endometrial)) OR (Endometrial Carcinomas)) OR (EC)) OR (Cancer, Endometrial)) OR (Cancers, Endometrial)) OR (ECs)) OR (Endometrium Cancer)) OR (Cancer, Endometrium)) OR (Cancers, Endometrium)) OR (Cancer of the Endometrium)) OR (Carcinoma of Endometrium)) OR (Endometrium Carcinoma)) OR (Endometrium Carcinomas)) OR (Cancer of Endometrium)) OR (Endometrium Cancers)) AND ((Monocyte) OR (MonocyteS))) AND ((((((Lymphocytes) OR (Lymphocyte)) OR (Lymphoid Cells)) OR (Cell, Lymphoid)) OR (Cells, Lymphoid)) OR (Lymphoid Cell))) AND (ratio) (Search term interoperability in databases). The literature screening process is shown in Fig. 1.

### Selection criteria

Eligible studies meeting the following criteria were included: (1) cohort studies including EC patients diagnosed by histopathology, (2) pretreatment cutoff values of LMR, (3) a hazard ratio (HR) with corresponding 95% confidence interval (CI), and (4) reporting outcome indicators, such as overall survival (OS), progression-free survival (PFS), disease-free survival (DFS), or cancer-specific survival (CSS). Articles were excluded for (1) insufficient data for HR and 95% CI, (2) unavailable full text, (3) conference abstract, cellular experiment, animal research, and non-English articles. Additionally, the reference lists of the retrieved articles were reviewed manually to identify additional relevant articles. The literature screening was conducted by two independent reviewers (Z. Huang and D. Yang) based on titles and abstracts initially. The remaining articles were further screened by full-text assessment. Any disagreements between the two reviewers on literature screening were resolved by discussion with a third author (C. Liu) to reach a consensus.

### Data extraction

Two reviewers (Z. Huang and D. Yang) independently extracted data from each study, including first author, country, publication year, sample size, age, BMI, FIGO staging, treatment method, cutoff value of LMR, and HR and 95% CI of OS, PFS, DFS, or CSS. Detailed information is provided in Table 1.

### Quality assessment

Study quality was appraised by two independent investigators (Z. Huang and D. Yang) using the Newcastle Ottawa Scale (NOS) (17). A NOS score of $\geq 6$ implied high quality.

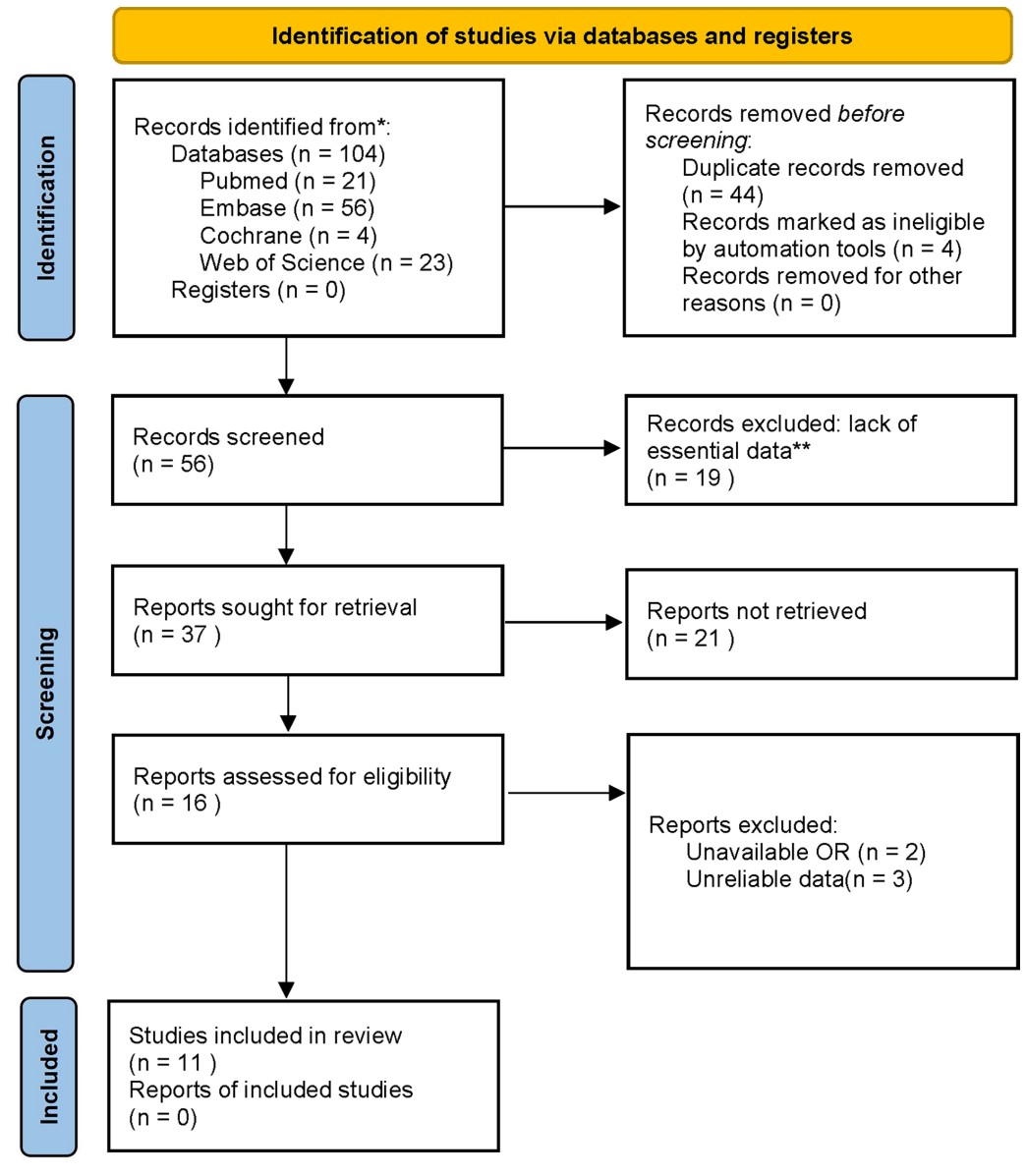

**Figure 1  The literature screening process.**

## Statistical analysis

The association between LMR and OS, DFS, and CSS was assessed using HR and 95% CI. Heterogeneity was judged using Cochran's $Q$ test and Higgins $I^2$ statistic. A random-effects model was adopted when $I^2 > 50\%$. Subgroup analyses were further performed by race (Asian or non-Asian), sample (<400 or $\geq$400), and age (<60 or $\geq$60) to evaluate the consistency of the prognostic value of LMR across diverse population groups and to determine the effect of sample size on the prognostic utility of LMR. Detailed information is provided in Table 2. In addition, sensitivity analysis was conducted to evaluate the effect of single studies on the pooled results for outcomes with marked heterogeneity. Publication

bias was determined *via* Egger's test, with $P < 0.05$ indicating statistical significance. All statistical analyses were done using STATA 15.1 software.

## RESULTS

### Prognostic value of LMR for OS

Eight studies explored the association between LMR and OS, involving 1,997 patients. Our results noted that LMR was correlated with OS (HR, 2.29; 95% CI [1.50–3.51]; $P = 0.001$) (Fig. 2). Subgroup analyses unveiled that this correlation was not influenced by race (Asian: HR, 3.21; 95% CI [1.52–6.77]; $P = 0.002$; non-Asian: HR, 1.64; 95% CI [1.00–6.71]; $P = 0.05$), median age (<60: HR, 3.52; 95% CI [1.28–9.68]; $P = 0.01$; ≥60: HR, 1.84; 95% CI [1.15–2.94]; $P = 0.01$), sample size (≥400: HR, 1.53; 95% CI [1.23–1.90]; $P = 0.0001$; <400: HR, 5.97; 95% CI [2.70–13.22]; $P = 0.0001$), and LMR cutoff (<5: HR, 2.91; 95% CI [1.41–5.98]; $P = 0.004$; ≥5: HR, 2.00; 95% CI [1.07–3.73]; $P = 0.03$).

### Prognostic value of LMR for DFS

Five studies with 1,590 patients assessed the link between LMR and DFS. Our results unraveled that LMR was correlated with DFS (HR, 4.00; 95% CI [1.76–9.07]; $P = 0.009$). Subgroup analyses evinced that this correlation was not influenced by race (Asian: HR, 5.04; 95% CI [1.51–16.84]; $P = 0.009$; non-Asian: HR, 2.29; 95% CI [1.41–3.72]; $P = 0.0008$) and LMR cutoff (<5: HR, 4.26; 95% CI [1.48–12.24]; $P = 0.007$; ≥5: HR, 3.65; 95% CI [1.60–8.31]; $P = 0.002$). However, this correlation was influenced by median age (<60: HR, 5.93; 95% CI [0.93–37.71]; $p = 0.06$; ≥60: HR, 2.58; 95% CI [1.70–3.92]; $P = 0.00001$) and sample size (≥400: HR, 1.75; 95% CI [0.95–3.22]; $P = 0.07$; <400: HR, 8.26; 95% CI [2.95–23.13]; $P = 0.0001$) (Figs. 3 and 4).

### Predictive value of LMR for CSS

Three studies with 1,282 patients estimated the correlation between LMR and CSS. Our results manifested that LMR was connected with CSS (HR, 1.58; 95% CI [1.11–2.25]; $P = 0.01$) (Fig. 5).

### Sensitivity analysis

Sensitivity analysis disclosed that the pooled OR was not changed for OS and DFS after exclusion of any individual study (Figs. 6–7).

### Publication bias

Eight articles on the link between LMR and OS were enrolled. Funnel plots signaled that LMR and OS were roughly symmetrical (Figs. 8–9), with a low probability of publication bias. Consistently, the Egger's test showed no publication bias ($P = 0.015$).

Five articles on the link between LMR and DFS were enrolled. Funnel plots manifested that LMR and DFS were roughly symmetrical, with a low probability of publication bias. Consistently, the Egger's test showed no publication bias ($P = 0.105 > 0.05$).

## Table 1   Basic characteristics.

| Author | Study period | Region | Study design | Timing | No. of patients | Age | BMI | TNM stage | LMR threshold | CSS | OS | DFS | Multivariate Cox regression |
|---|---|---|---|---|---|---|---|---|---|---|---|---|---|
| Cummings et al., 2015 | 2005–2007 | UK | Retrospective cohort | Surgically treated | 605 | 65 (28–95) | N/A | I–IV | 5.26 | 1.26 (0.73–2.15) | 1.23 (0.84–1.82) | N/A | Combined NLR+PLR, age, stage, grade, histopathological subtype, and lymphvascular space invasion |
| Eo et al., 2016 | 2005–2014 | Korea | Retrospective cohort | Hysterectomy-based comprehensive surgical staging | 255 | 44 (28–82) | N/A | I–IV | 3.28 | N/A | 0.07 (0.02, 0.24) | 0.10 (0.03, 0.32) | Histological grade, FIGO stage, LMR |
| Cömert et al., 2018 | 2005–2016 | Turkey | Retrospective cohort | At least total abdominal hysterectomy and bilateral salpingo-oophorectomy | 497 | 58 (29–92) | N/A | I–IV | 5.46 | N/A | 1.66 (0.64–4.29) | 1.22 (0.62–2.38) | Stage, Platelet-to-lymphocyte ratio, and [Platelet plus Neutrophil plus Monocyte]-to-lymphocyte ratio |
| Cong et al., 2020 | 2013–2017 | China | Retrospective cohort | Hysterectomy (with or without adnexectomy and lymphadenectomy) | 1,111 | 56 | N/A | I–IV | 4.55 | N/A | 1.72 (1.20–2.45) | N/A | Age, stage, grade, lymphovascular space invasion, histopathological subtype, NLR, and PLR |
| Holub et al., 2020 | 2008–2017 | France&Spain | Retrospective cohort | Postoperative External Beam Radiotherapy (EBRT) | 155 | 63.1 (27.9–98.9) | N/A | I–III | 5.56 | 5.4 (1.3–22.7) | 5.9 (1.4–24.6) | N/A | FIGO stage and lymphocytes |
| Cubukcu et al., 2021 | 2010–2019 | Turkey | Retrospective cohort | Postsurgical; FIGO stage I disease with histological endometrioid type and grade 1 or 2, invading less than onehalf of the myometrium without lymphovascular space invasion (LVSI) | 253 | 58.5 (32.0–75.4) | N/A | I | 4.71 | N/A | N/A | N/A | NLR and Ki-67 index |
| Song et al., 2021 | 2010–2019 | Korea | Retrospective cohort | Primary surgical treatment including total hysterectomy, bilateral salpingo-oophorectomy, and systematic lymphadenectomy; adjuvant chemotherapy, radiation therapy, or a combination of chemotherapy and radiation | 118 | 61(42–83) | N/A | I–IV | 5.24 | N/A | 2.941 (1.210–7.147) | 3.647(1.600–8.315) | FIGO stage and LMR |
| Ahn et al., 2022 | 2010–2020 | Korea | Retrospective cohort | Primary surgical treatment | 225 | 54 (28–81) | N/A | I | 4.55 | N/A | N/A | 20.643 (5.616–75.873) | Grade, MMI proportion, adjuvant radiotherapy, and MLR |
| Bizzarri et al., 2022 | 2013–2019 | Italy | Retrospective cohort | Primary surgery | 495 | 63 (26–88) | N/A | I–IV | 3.33 | N/A | N/A | 2.288 (1.409–3.716) L/H | Molecular and systemic inflammatory markers |
| Njoku et al., 2022 | 2010–2015 | UK | Retrospective cohort | N/A | 522 | 66 (56–73) | N/A | I–IV | 4.00 | 1.64 (1.01–2.67) | 1.66 (1.09–2.50) | N/A | age, BMI, histology, grade, FIGO stage, LVSI, depth ofmyometrial invasion, T2DM status and treatment received. |
| Bing, Tsui & Ding, 2022 | 2011–2021 | Taiwan, China | Retrospective cohort | Hysterectomy-based surgery | 48 | 56.77 ±8.79 | N/A | I–IV | 4.19 | N/A | 8.88 (1.03, 76.28) | N/A | Age, DM, and MLR |

**Table 2  Subgroup analysis.**

| Subgroup | OS | | | | DFS | | | |
|---|---|---|---|---|---|---|---|---|
| | Study | HR (95% CI) | *P* value | $I^2$ | Study | HR (95% CI) | *P* value | $I^2$ |
| Total | 8 | 2.29 [1.50–3.51] | 0.004 | 66% | 5 | 4.00 [1.76–9.07] | 0.0009 | 80% |
| sample size | | | | | | | | |
| >400 | 4 | 1.53 [1.23–1.90] | 0.0001 | 0% | 2 | 1.75 [0.95–3.22] | 0.07 | 54% |
| ≤400 | 4 | 5.97 [2.70–13.22] | 0.0001 | 32% | 3 | 8.26 [2.95–23.13] | 0.0001 | 63% |
| Median age | | | | | | | | |
| >60 yrs | 4 | 1.84 [1.15–2.94] | 0.01 | 56% | 2 | 2.58 [1.70–3.92] | 0.00001 | 0% |
| ≤60 yrs | 4 | 3.52 [1.28–9.68] | 0.01 | 76% | 3 | 5.93 [0.93–37.71] | 0.06 | 90% |
| Region | | | | | | | | |
| Asia | 5 | 3.21 [1.52–6.77] | 0.002 | 69% | 4 | 5.04 [1.51–16.84] | 0.009 | 85% |
| Europe | 3 | 1.64 [1.00-6.71] | 0.05 | 58% | 1 | 2.29 [1.41–3.72] | 0.0008 | |
| LMR | | | | | | | | |
| >5 | 4 | 2.00 [1.07–3.73] | 0.03 | 56% | 1 | 3.65 [1.60–8.31] | 0.002 | |
| ≤5 | 4 | 2.91 [1.41–5.98] | 0.004 | 77% | 4 | 4.26 [1.48–12.24] | 0.007 | 85% |

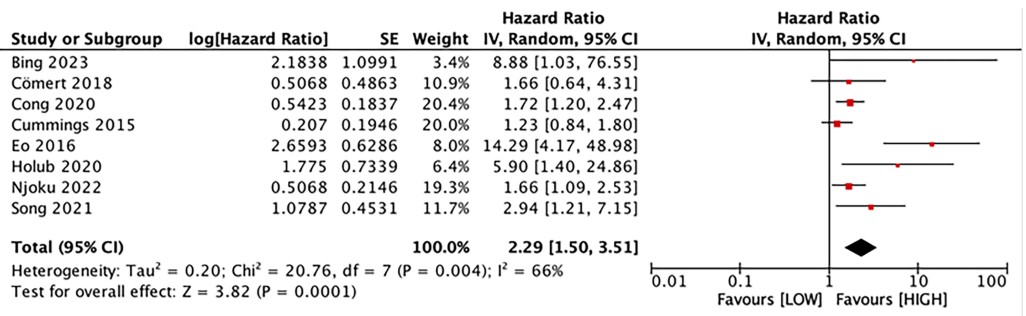

**Figure 2  Forest plot of the association between LMR and OS.** Studies: *Bing, Tsui & Ding, 2022*; *Cömert et al., 2018*; *Cong et al., 2020*; *Cummings et al., 2015*; *Eo et al., 2016*; *Holub et al., 2020*; *Njoku et al., 2022*; *Song et al., 2021*.

**Figure 3  Forest plot of the association between LMR and DFS.** Studies: *Ahn et al., 2022*; *Bizzarri et al., 2022*; *Cömert et al., 2018*; *Eo et al., 2016*; *Song et al., 2021*.

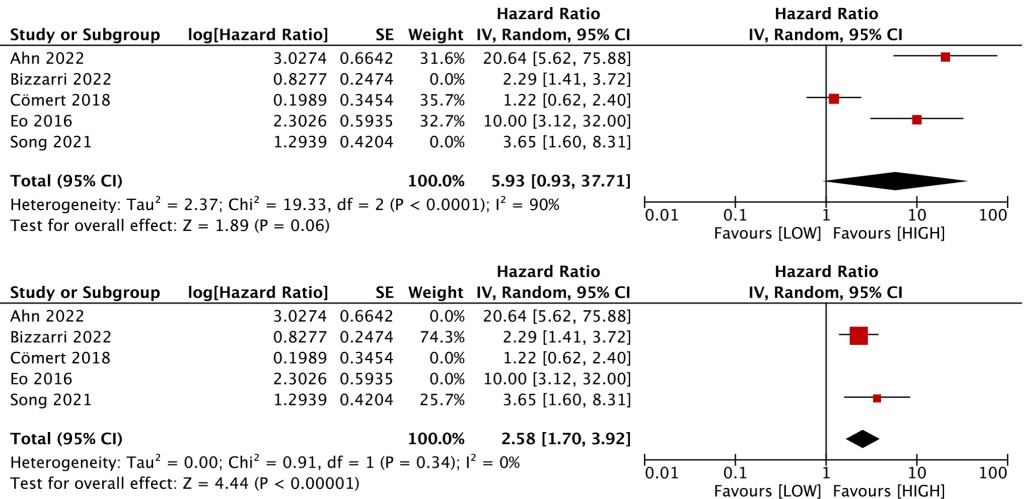

**Figure 4  Subgroup analysis on median age of the association between LMR and DFS.** Studies: *Ahn et al., 2022*; *Bizzarri et al., 2022*; *Cömert et al., 2018*; *Eo et al., 2016*; *Song et al., 2021*.

**Figure 5  Forest plot of the association between LMR and CSS.** Studies: *Cummings et al., 2015*; *Holub et al., 2020*; *Njoku et al., 2022*.

# DISCUSSION

Precise assessment of metastasis risk at the time of diagnosis and the risk of future recurrence is essential for formulating individualized treatment plans for cancer patients, improving prognosis, and minimizing the side effects of unnecessary adjuvant therapy. In recent years, new methods have emerged to classify EC patients into low-, intermediate-, or high-risk groups. Lymph node metastasis and lymph-vascular space invasion are also critical in the risk stratification of these patients (*Kasius et al., 2021*). The latest FIGO staging of EC in 2023 also incorporates molecular classification from The Cancer Genome Atlas (TCGA), a comprehensive project that maps genomic changes in various cancers.

Numerous efforts have been made to enhance the care quality for EC women. According to *Ahn et al. (2022)*, a substantial number of low-risk EC patients have encountered recurrence, which cannot be predicted by traditional parameters. Therefore, it is imperative to identify new indicators for the early detection of possible recurrences. A previous study discovered that LMR was a marker for DFS (*Concin et al., 2021*). Our aggregated data on EC prognosis unraveled that a lower LMR was correlated with decreased OS and DFS.

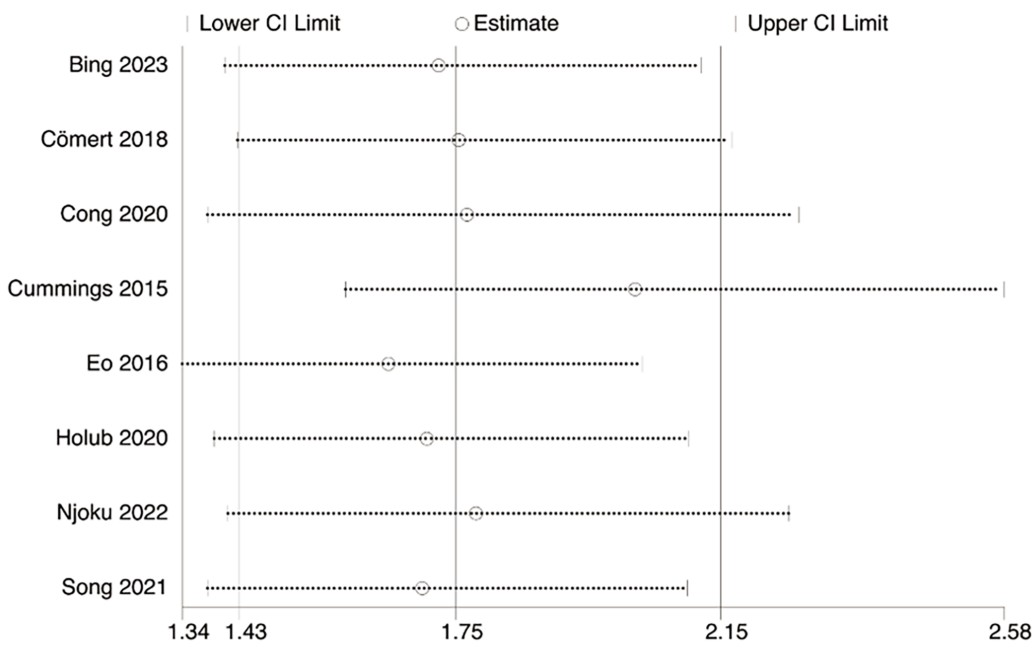

**Figure 6** **Sensitivity analysis of OS.** Studies: *Bing, Tsui & Ding, 2022*; *Cömert et al., 2018*; *Cong et al., 2020*; *Cummings et al., 2015*; *Eo et al., 2016*; *Holub et al., 2020*; *Njoku et al., 2022*; *Song et al., 2021*.

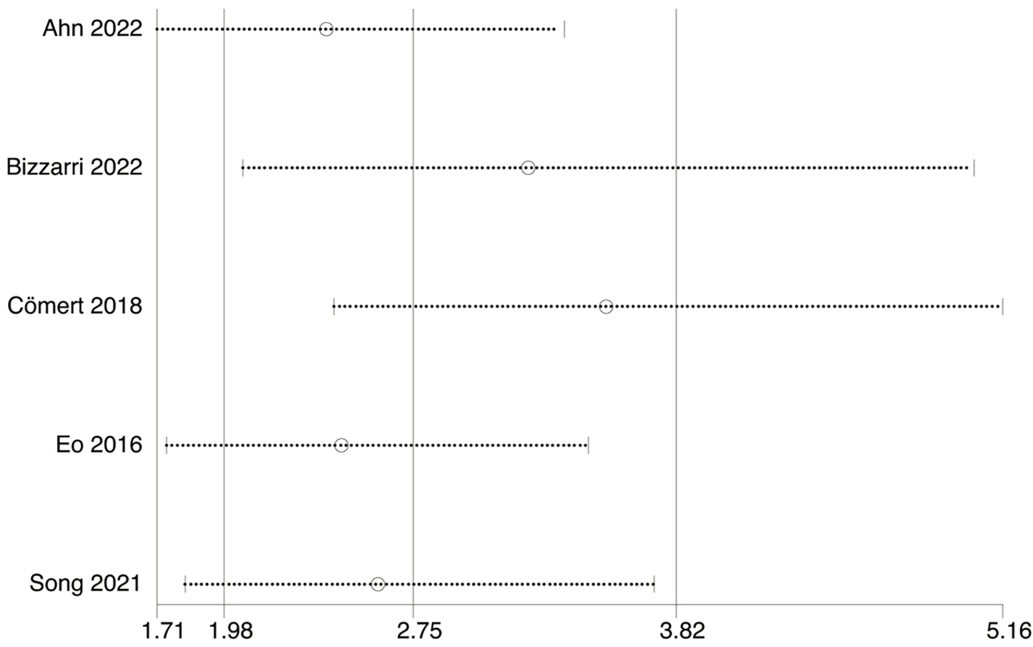

**Figure 7** **Sensitivity analysis of DFS.** Studies: *Ahn et al., 2022*; *Bizzarri et al., 2022*; *Cömert et al., 2018*; *Eo et al., 2016*; *Song et al., 2021*.

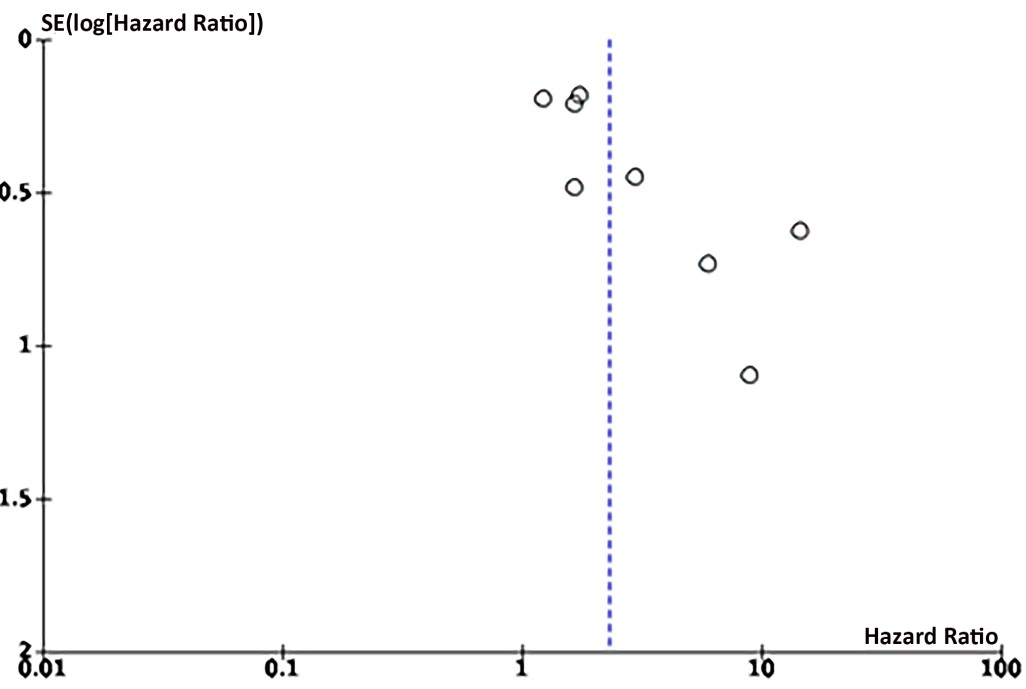

**Figure 8** Funnel plot of OS.

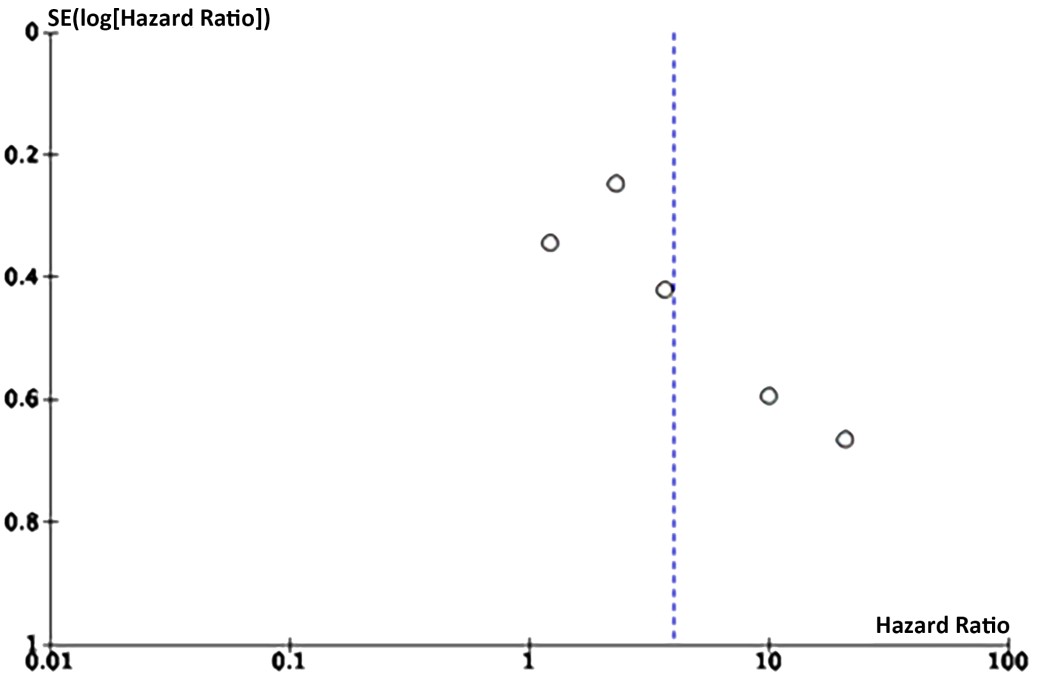

**Figure 9** Funnel plot of DFS.

Sensitivity analysis and subgroup analyses further confirmed the significant predictive value of LMR for EC prognosis.

Biological indicators that are associated with tumor inflammation and the surrounding microenvironment can serve as significant prognostic markers for cancer patients. The link between inflammation and cancer was initially elucidated by Virchow in 1863, and subsequent studies unveiled the specific functions of immune cells, notably lymphocytes and monocytes (*Balkwill & Mantovani, 2001*; *Coussens & Werb, 2002*; *Grivennikov, Greten & Karin, 2010*). Monocytes play diverse roles in the anti-cancer immune response. They can differentiate into tumor-associated macrophages, which may either drive tumor progression by promoting angiogenesis and metastasis and suppressing anti-tumor immune responses or exert anti-tumoral effects by phagocytosing tumor cells and producing cytotoxic molecules (*Chapman et al., 2004*; *Olingy, Dinh & Hedrick, 2019*). Recent studies have highlighted the critical role of peripheral blood mononuclear cells (PBMCs) in the prognosis of EC. Studies reveal their dual impact through biomarker regulation and immune microenvironment modulation. Elevated monocyte-associated KIF2C correlates with poor EC outcomes by suppressing CD8+ T cell infiltration (*An et al., 2021*). The Pan-Immune-Inflammation Value (PIV), incorporating monocyte parameters, predicts survival with higher levels indicating worse prognosis (*Guven et al., 2022*). Monocyte dynamics influence immunotherapy efficacy, particularly in POLE/MSI-H subtypes *via* PD-1/PD-L1 axis regulation (*Cao et al., 2021*). Emerging strategies combine monocyte depletion/phenotype reprogramming with vaccines or cell therapies to enhance T cell responses. These findings position monocytes as both prognostic biomarkers and immune system integrators.

Conversely, lymphocytes are primarily acknowledged for their anti-tumor effects, such as inducing apoptosis and inhibiting proliferation. CD8+ T lymphocytes are known for their cytotoxic effects against tumor cells, whereas CD4+ T lymphocytes are recognized for their robust anti-tumor immune responses (*Huang et al., 2022*; *Takeuchi & Saito, 2017*; *Tsukumo & Yasutomo, 2018*). Recent studies have indicated a correlation between heightened monocyte counts and elevated levels of circulating pro-inflammatory mediators, which fosters a conducive environment for the proliferation of cancer cells (*Stiekema et al., 2020*; *Zingaropoli et al., 2021*). LMR holds promise for guiding novel approaches in cancer prevention and treatment, making it a crucial factor for predicting prognosis in various malignancies. However, the exact mechanisms behind the link between elevated LMR and adverse outcomes are not yet fully understood, although it reflects the detrimental function of monocytes and the beneficial prognostic implication of lymphocytes (*Camerino et al., 2021*; *Trova et al., 2019*; *Zhu et al., 2022*; *Zuo et al., 2023*).

By integrating the latest data, our findings aligned with prior research, suggesting an association between LMR and survival outcomes in EC patients. Following surgical intervention, it is crucial to conduct an accurate prognostic assessment and develop individualized therapeutic strategies, regardless of disease stage. This approach mitigates the risk of undertreatment, thereby reducing the risk of tumor recurrence, which poses a significant threat to patient survival and overall prognosis (*Lin et al., 2021*; *Zhang & Li, 2023*). Moreover, it minimizes the risk of overtreatment-related complications in adjacent

organs, particularly in radiotherapy, where precise dosage and proximity effect are essential (*Allen & Daescu, 2013*). Additionally, a higher LMR may help predict EC prognosis and guide surgical decisions, suggesting more aggressive treatment strategies for those with elevated levels. It serves as a convenient and accessible biomarker that complements these methods by facilitating the precise identification of patients with poor prognosis and providing a novel tool for constructing predictive models and identifying potential research targets.

Subgroup analyses revealed that prognostic prediction for EC recurrence was more applicable to patients aged over 60 years. Given the rising incidence of EC in developing nations, additional investigations in diverse geographical regions and younger patients (*Sajo et al., 2020*; *Siegel et al., 2022*) are warranted. Such endeavors would significantly enhance global efforts for EC prevention and help clinically resolve atypical cases by referencing these variables.

However, our meta-analysis presents several limitations. First, all enrolled studies were retrospective observational studies, indicating that our results relied on unadjusted estimates. The precision could be enhanced by considering additional confounding variables such as age, BMI, and lifestyle. Second, the analysis was limited to just eight eligible studies, encompassing 4,284 patients, which may not provide a robust foundation for comprehensive analysis, particularly in subgroup analyses. The sample size may not adequately ensure the stability of results or support comprehensive subgroup analyses, potentially leading to publication bias despite no statistical evidence thereof. Third, the study populations were predominantly Asians, limiting the generalizability of our findings to a global population. Furthermore, additional investigations are warranted on other parameters, such as the platelet-to-lymphocyte ratio (PLR) and neutrophil-to-lymphocyte ratio (NLR), which have been discussed in several studies (*Bing, Tsui & Ding, 2022*; *Ahn et al., 2022*; *Njoku et al., 2022*). Further research on these markers is essential to provide a more comprehensive understanding of their prognostic implications in EC.

## CONCLUSIONS

In summary, our study reveals a significant association between elevated LMR and improved OS and DFS in postoperative EC patients. These findings suggest that LMR is a reliable and practical predictive indicator for postoperative EC prognosis. Nonetheless, further prospective investigations are warranted to validate our results and ascertain appropriate cut-off values.

### Funding

This study was supported by the National Natural Science Foundation of China (Grant number 82072882). The funders had no role in study design, data collection and analysis, decision to publish, or preparation of the manuscript.

## Grant Disclosures

The following grant information was disclosed by the authors:
National Natural Science Foundation of China: 82072882.

## Competing Interests

The authors declare there are no competing interests.

## Author Contributions

- Zijing Huang conceived and designed the experiments, performed the experiments, analyzed the data, prepared figures and/or tables, authored or reviewed drafts of the article, and approved the final draft.
- Donghua Yang performed the experiments, analyzed the data, prepared figures and/or tables, authored or reviewed drafts of the article, and approved the final draft.
- Congrong Liu conceived and designed the experiments, authored or reviewed drafts of the article, funding acquisition, and approved the final draft.

## Data Availability

   This is a systematic review/meta-analysis.

## Supplemental Information

Supplemental information for this article can be found online at http://dx.doi.org/10.7717/peerj.19345#supplemental-information.

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
