# Peer review of "Prognostic value of lymphocyte-to-monocyte ratio in patients with endometrial cancer: an updated systematic review and meta-analysis"

_PeerJ, doi:10.7717/peerj.19345_

## Round 0.1 · original submission · Major Revisions

The authors are requested to carefully revise the manuscript and answer the questions raised by the reviewers.

Reviewer 1 ·

Basic reporting

The authors conducted a meta-analysis on the prognostic value of lymphocyte-to-monocyte ratios in endometrial cancer. The results are clear and well described. While the paper is well-written, it requires some corrections in English wording before publication in PeerJ.

Experimental design

The clinical questions are relevant, and the experimental design is well-developed. However, it is not clear at the moment why the LMR was chosen.
Factors affecting LMR have been examined, but it is equally important to consider factors other than LMR that impact OS and DFS.

Validity of the findings

To make the conclusions more robust, an extensive examinations of factors related to DFS, aside from LMR, is necessary.

Additional comments

Minor comments.
1.Some sentences are quite complex and should be simplified or split into smaller parts for better clarity. English editing is expected to make the paper easier for readers to understand.

The description of the results is difficult to follow as it focuses solely on data. Additionally, the lack of line breaks in the discussion section makes it hard to read, so the presentation needs improvement.

2. What is CSS?

Reviewer 2 ·

Basic reporting

Huang et al present a manuscript with meta-analysis that highlights LMR as a prognostic biomarker for OS and DFS in EC patients. . They showed that LMR was significantly associated with OS and PFS.

Experimental design

After curation, authors included 8 studies for OS and 5 studies for PFS in the meta analysis that met the criteria laid out by the authors. They compared LMR with OS and PFS.

Validity of the findings

Figure 8 and 9 is hard to read,

I have following questions about the findings:

1. Does the ratio correlate with other known prognostic markers? ECOG normalization?
2. How does the Genomic profile of LMR low vs high group compare?
3. How was high/low LMR defined?
4. What would be the recommended LMR cutoff to decide course of action for EC patients?

Reviewer 3 ·

Basic reporting

a. The conclusions drawn in the abstract seem out of proportion to the results. In lines 45-50 it is implied that LMR could affect treatment planning and improve outcomes through personalized treatment approaches. While LMR may be prognostic, we don’t have data to suggest that LMR should alter a patient’s treatment plan. I would consider re-wording this to focus just on the prognostic implications.
b. Line 58 of the introduction states that the exact etiology of endometrial cancer is unknown, however we do understand the etiology in a large number of cases (estrogen-mediated, genetic, etc).
c. When discussing risk-stratification of endometrial cancer, it is important to mention traditional pathologic features (depth of myometrial invasion, lymphovascular space invasion, histology/grade). It would also be important to comment upon molecular classification as reported by the TCGA/ProMisE, as this has been shown to significantly improve risk-stratification and is now incorporated into staging and treatment guidelines. I would add these into the first paragraph of the introduction (before line 62).
d. I do not feel it is imperative that you include the information about LMR and association with cardiovascular disease (lines 70-74).
e. The wording in lines 75-79 is very confusing. Suggesting that LMR is prognostic would imply that it is associated with survival outcomes, whereas suggesting that LMR is predictive of treatment response would be a different statement indicating that LMR is able to predict how a patient will respond to a particular therapy. Please clarify these sentences (would not use prognostic and predictive in the same sentence).

Experimental design

a. Line 102: please specify what the hazard ratio was for – overall survival, disease specific survival, or both?
b. Line 102: this states that pretreatment cutoff values for LMR were included. What does “pretreatment” refer to? Were included studies evaluating patients at the time of surgical management of endometrial cancer? Adjuvant chemotherapy? Adjuvant radiation therapy? Recurrent setting? I anticipate that the prognostic value of LMR would be significantly different across these different groups of patients, and this would significantly affect the results of the study.
c. Line 104-105: “(5) cohort studies” should be moved to the beginning of this paragraph (may be included with 1: EC patients diagnosed by histopathology).

Validity of the findings

a. Line 135: would change to say “prognostic value of LMR for OS”
b. Line 143: would change to say “prognostic value of LMR for DFS”
c. Line 152: please define CSS
d. In the results section, please expand upon whether studies included patients who underwent surgical management of endometrial cancer, adjuvant treatment (chemotherapy and/or radiation therapy), or recurrent treatment. I would either recommend separating these and evaluating the LMR for each as a distinct group (again, it is possible that the prognostic value of LMR is different for each), OR control for this in your subgroup/multivariable analysis for OS, DFS, CSS.
e. Line 138-142: In the subgroup analysis you state that the correlation was not influenced by race, however the HR for Asian was statistically significant (p=0.01), and non-Asian was NOT statistically significant (p=0.09). The same is true for age (one p-value significant, the other is not). For sample size, neither p-value is significant. Please elaborate on how you came to your conclusion that LMR correlation was not influenced by these factors, as the statistical analysis does not seem to support this.
f. Line 146-148: same as above. You state that the subgroup analysis for LMR correlation with DFS was not influenced by race or sample size, yet the variation of p-values do not seem to support this.
g. Lines 172-173: FIGO stage and lymph node metastasis imply the same concept of advanced stage/metastatic disease, thus both do not need to be stated.
h. Line 175: please define TCGA.
i. Line 204: by MLR, do you mean LMR?
j. Line 216: please clarify what you mean by “there is potential to predict EC onset” – to me, this implies that LMR could predict endometrial cancer before it occurs, which is not supported by your study results.
k. Line 216-217: please expand upon the statement made that LMR may facilitate surgical decision-making and treatment efficacy. How/why would this change the intra-operative management or decision for adjuvant treatment?
l. Line 218: in your analysis you looked at Asian vs non-Asian, but I would not jump to the conclusion that the results were more applicable for the European healthcare system.
m. Line 221-223: your study results do provide data that would support endometrial cancer prevention or resolve rare cases, so I would not make this conclusion.

Additional comments

a. This is an important concept to report on, and I applaud the authors for doing so.
b. In the introduction section, I would expand upon current risk-stratification protocols for endometrial cancer (traditional including stage, myometrial invasion, lymphovascular space invasion, histology/grade), and molecular classification. In the discussion, I would significantly expand upon how LMR may fit into these pre-existing risk-stratification algorithms in order to enhance our understanding of endometrial cancer prognosis.
c. Please comment upon the treatment setting of these patients in both the results as well as the discussion – were they undergoing surgical evaluation/staging, systemic chemotherapy (upfront vs recurrence), radiation therapy, targeted therapy, some combination of the above? Is LMR useful in all of these circumstances?

---

## Round 0.2 · Major Revisions

The authors are requested to carefully revise the manuscript and answer the questions raised by the reviewers.

Reviewer 1 ·

Basic reporting

The revision has made the manuscript more readable. However, it remains unclear whether LMR is indeed more useful than other prognostic indicators.

The authors suggest that LMR is associated with prognosis based on factors such as age, but is LMR truly more significant than traditional staging systems?

Although the study includes a large number of cases, it is a retrospective analysis, and the treatments used may vary across cases. The rationale for treating these varying treatments as equivalent in the analysis remains unclear.

Experimental design

Are all treatments used in the 8 papers the same treatment?
Are they treated with the same criteria?

Validity of the findings

There are no major problems with the methodology.

Additional comments

Why is LMR related to prognosis? What is the underlying mechanism?

Reviewer 2 ·

Basic reporting

The authors made changes based on the suggestions made. I have no further comments.

Experimental design

The authors made changes based on the suggestions made. I have no further comments.

Validity of the findings

The authors made changes based on the suggestions made. I have no further comments.

Reviewer 4 ·

Basic reporting

clear

Experimental design

Potentially valid. Need more clarifications in methods to confirm.

Validity of the findings

Results need more clarification

Additional comments

Authors assessed prognostic value of lymphocyte-to-monocyte ratio (LMR) in patients with endometrial cancer by meta-analysis. Results showed higher LMR is associated with poorer overall survival (1,997 patients from 8 studies) and poorer disease-free survival (1,590 patients from 5 studies), and cancer-specific survival (1,282 patients from 3 studies). Previous reviewers have raised multiple valid points.
Some additional comments:
1. Please include acknowledged prognostic factors (per ECOG guideline) and treatment for each study as they determine if these cohorts are comparable to be included in this study. Also please summarize what covariates were adjusted in each study.
2. Can add a table listing key demographic and prognostic information of patients.
3. As mentioned by reviewer 2, the cut-off of low vs high LMR needs a clarification but that was not consistent across studies, which threatens validity of analysis and conclusion. Please clarify how the added subgroup analyses were performed and describe the results in more details. Also, was LMR used as numeric or categorical (high/low) throughout this study? Please clarify.
4. Need to add cancer-specific survival results into introduction.
5. Are there studies of PBMC / peripheral circulating immune system and their association in EC? Are there hypotheses why / how this information adds value to clinically acknowledged prognostic factors? Please add them into introduction / discussion.

---

## Round 0.3 · Major Revisions

The authors are requested to carefully revise the manuscript and answer the questions raised by the reviewers.

Reviewer 4 ·

Basic reporting

Please see additional comments

Experimental design

Please see additional comments

Validity of the findings

Please see additional comments

Additional comments

Authors assessed prognostic value of lymphocyte-to-monocyte ratio (LMR) in patients with endometrial cancer by meta-analysis. Results showed higher LMR is associated with poorer overall survival (1,997 patients from 8 studies) and poorer disease-free survival (1,590 patients from 5 studies), and cancer-specific survival (1,282 patients from 3 studies). Previous reviewers have raised multiple valid points.
Some additional comments:
1. Please include acknowledged prognostic factors (per ECOG guideline) and treatment for each study as they determine if these cohorts are comparable to be included in this study. Also please summarize what covariates were adjusted in each study.
Response: Thank you for your valuable comment. Unfortunately, the original studies did not provide complete ECOG data. We have added a discussion on the acknowledged prognostic factors and treatments mentioned in the studies, and their relevance to other cancer types. This information is summarized in Table 1 for clarity. However, due to limited data availability, relevant subgroup analyses could not be performed.
Yet, I did not see covariates adjustment information in Table 1 (eg, Cox regression performed adjusting for age, stage, etc.)

2. Can add a table listing key demographic and prognostic information of patients.
Response: Thank you for the suggestion. Key demographic and prognostic information of patients has been summarized in Table 1.
Addressed

3. As mentioned by reviewer 2, the cut-off of low vs high LMR needs a clarification but that was not consistent across studies, which threatens validity of analysis and conclusion. Please clarify how the added subgroup analyses were performed and describe the results in more details. Also, was LMR used as numeric or categorical (high/low) throughout this study? Please clarify.
Response: Thank you for your comment. For quantitative analysis, we conducted subgroup analyses, with detailed results presented in Table 2. This has been clarified in the revised manuscript.
This response did not answer the questions. Please show definition of LMR high / low in each source study.

4. Need to add cancer-specific survival results into introduction.
Response: Thank you for pointing this out. Cancer-specific survival results have been added to the Introduction section.
Addressed.

5. Are there studies of PBMC / peripheral circulating immune system and their association in EC? Are there hypotheses why / how this information adds value to clinically acknowledged prognostic factors? Please add them into introduction / discussion.
Response: Thank you for your thoughtful comment. The association between PBMC/peripheral circulating immune components and EC has already been addressed in the manuscript’s discussion of underlying mechanisms. Additionally, we have clarified the purpose and potential advantages of incorporating this information, emphasizing how it complements clinically acknowledged prognostic factors by offering insights into tumor-immune interactions and providing a convenient, non-invasive tool for prognosis.
The original discussion is very unspecific. Circulating immune system and TME are different, and that describing biology of lymphocytes and monocytes without a cancer-specific context is vague. Please use more specific and up-to-date literature of PMBC studies of EC.

---

## Round 0.4 · accepted · Accept

Although one reviewer previously recommended rejection, the other two reviewers have recommended acceptance. I also reviewed the manuscript and found no obvious risks to publication. Therefore, I also approved the publication of this manuscript.

Reviewer 4 ·

Basic reporting

See additional comments

Experimental design

See additional comments

Validity of the findings

See additional comments

Additional comments

Authors' updates for the previous round of review were acceptable.